# Papillary Thyroid Cancer Remodels the Genetic Information Processing Pathways

**DOI:** 10.3390/genes15050621

**Published:** 2024-05-14

**Authors:** Dumitru Andrei Iacobas, Sanda Iacobas

**Affiliations:** 1Personalized Genomics Laboratory, Undergraduate Medical Academy, Prairie View A&M University, Prairie View, TX 77446, USA; 2Department of Pathology, New York Medical College, Valhalla, NY 10595, USA; sandaiacobas@gmail.com

**Keywords:** 8505C anaplastic thyroid cancer cell line, BCPAP papillary thyroid cancer cell line, DNA replication, DNA repair, DNA transcription, evading apoptosis, proliferation, RNA polymerase, TATA-box binding protein associated factors, UBXN1

## Abstract

The genetic causes of the differentiated, highly treatable, and mostly non-fatal papillary thyroid cancer (PTC) are not yet fully understood. The mostly accepted PTC etiology blames the altered sequence or/and expression level of certain biomarker genes. However, tumor heterogeneity and the patient’s unique set of favoring factors question the fit-for-all gene biomarkers. Publicly accessible gene expression profiles of the cancer nodule and the surrounding normal tissue from a surgically removed PTC tumor were re-analyzed to determine the cancer-induced alterations of the genomic fabrics responsible for major functional pathways. Tumor data were compared with those of standard papillary and anaplastic thyroid cancer cell lines. We found that PTC regulated numerous genes associated with DNA replication, repair, and transcription. Results further indicated that changes of the gene networking in functional pathways and the homeostatic control of transcript abundances also had major contributions to the PTC phenotype occurrence. The purpose to proliferate and invade the entire gland may explain the substantial transcriptomic differences we detected between the cells of the cancer nodule and those spread in homo-cellular cultures (where they need only to survive). In conclusion, the PTC etiology should include the complex molecular mechanisms involved in the remodeling of the genetic information processing pathways.

## 1. Introduction

The American Cancer Society estimates that in 2024, the USA will register 44,020 (31,520 women and 12,500 men) new cases of thyroid cancer (TC), out of which 2170 (1180 females + 990 males) might die because of it [1]. The most lethal form is the non-differentiated (anaplastic) TC, with only 5 months median survival rate [2] and 8% average 5-year Relative Survival Rate (RSR) when all Surveillance, Epidemiology, and End Results (SEER) stages are considered [1]. The least lethal is the differentiated papillary thyroid carcinoma (PTC) that covers over 8 out of 10 thyroid malignancies, grows slowly, and has a favorable (>99%) 5-year survival prognosis [3,4]. Non-lethal and treatable cancers also include follicular thyroid cancer (RSR = 98%) and medullary thyroid cancer (RSR = 91%). However, if not treated in time, the differentiated forms can invade adjacent structures and metastasize in neck lymph nodes [5,6] and develop aggressive variants [7,8]. 

It is considered that the T1799A point mutation of the *BRAF* gene is responsible for 45% of the PTC cases [9], albeit mutations of several other genes (e.g.,: *RET*, *NTRK1*, and *TP53*) were also reported, e.g., [10,11,12,13,14] were present in numerous PTC cases. In total, the 39.0 Release of the National Cancer Institute GDC Data Portal (4 December 2023) lists 5869 mutations affecting 19,313 genes in the registered 1121 cases of thyroid cancer [15]. In addition to specific DNA mutations, regulation of the expression levels of selected gene sets were also pointed out as PTC transcriptomic signatures (e.g., [16,17,18]). 

In order to quantify the power of the expression of a particular gene to regulate the cancer phenotype, we have introduced the Gene Commanding Height (GCH) score [19]. Nonetheless, our PTC transcriptomic study [20] revealed that 1/3 of the 544 known gene cancer biomarkers had higher GCH in the malignant region of the tumor and 1/3 in the benign region, while the rest did not discriminate between the two regions. This result raised the general question of the gene biomarkers’ utility for cancer diagnosis [21] and therapy [22], especially because, together with the blamed biomarker(s), the sequence and/or expression of numerous other protein-coding (e.g., [23]) and non-coding (e.g., [24]) genes are altered in the thyroid cancer [25]. Moreover, all our genomics studies on prostate (e.g., [19]), thyroid (e.g., [25]), and kidney (e.g., [26]) cancers revealed that biomarkers are far below the top GCH scorers, meaning that they are minor players in the cell life. Therefore, manipulation of their sequence or/and expression level might be of little consequences for the survival and proliferation of the malignancy.

Nonetheless, the blamed genes are only part of a larger category of medical signs that are considered to be indicative of cancer physiopathology and are used in the management of thyroid cancers [27]. The U.S. Food and Drug Administration (FDA), National Institutes of Health (NIH), and the European Medicines Agency (EMA) have defined what should be included among biomarkers [28,29]. 

Total thyroidectomy is the primary approach of treating PTC, especially with multi-focal occurrence, but lobectomy is less risky in papillary thyroid microcarcinomas [30]. The thyroid surgical approach is frequently combined with 131I radiotherapy (recommended in 2015 by the American Thyroid Association [31]) or with the administration of antiangiogenic multikinase inhibitors (e.g., lenvatinib, sorafenib, and cabozantinib) for cancers resistant to radioactive iodine [32]. Novel specific kinase inhibitors (SKI) like trametinib are also considered efficient for thyroid cancer treatment [33]. 

Here, we present the PTC-induced changes in the KEGG-constructed functional pathways [34] responsible for the genetic information processing in a surgically removed PTC tumor. The study is an addition to the not-so-rich literature related to the genomic alterations in DNA replication [35], repair [36,37,38,39] and transcription [40,41], chromatin remodeling [42], translation, and protein processing [43] and trafficking [44]. 

The analyses were carried out from the Genomic Fabric Paradigm (GFP) perspective which provides the most theoretically possible characterization of the transcriptome [45] and is able to identify the most legitimate targets for personalized anti-cancer gene therapy [46]. In addition to the traditional average expression level (AVE), GFP characterizes each gene with the Relative Expression Variation (REV) across biological replicas and Expression Coordination (COR), with each other gene in the same condition. 

By comparing AVEs in the cancer nodule and the surrounding normal thyroid tissue, one determines which gene was significantly up-/down-regulated according to certain cut-off criteria for the absolute fold-change and *p*-value of the *t*-test of means equality. AVE analysis, mandatory in all gene expression studies, is also used to identify the turned on/off genes with respect to the background-related detection limit of the platform. 

REV analysis identifies the genes whose random fluctuations of the expression levels caused by the stochastic nature of the transcription chemical reactions are the most/least controlled by the homeostatic mechanisms. Indirectly, REV points out the cell priorities by keeping, within narrow intervals, the expression levels of critical genes to preserve the cell phenotype under the pressure of the nonhomogeneous and variable local environmental factors. 

COR analysis is based on the Principle of Transcriptomic Stoichiometry [47] (an extension of the law of multiple proportions from chemistry), stating that genes whose encoded products interact should be coordinately expressed to maximize the pathway efficiency. Thus, COR analysis singles out the most probable, statistically (*p* < 0.05) significant gene networking in functional pathways, and allows for the quantification of the gene network remodeling in disease and its recovery following a treatment. 

## 2. Materials and Methods

### 2.1. Gene Expression Data

Transcriptomic raw data obtained through our previous microarray experiments on thyroid cancer samples were downloaded from the publicly available Gene Expression Omnibus (GEO) of the National Center for Biotechnology Information (NCBI). As described in [20], 4 small pieces of malignant (hereafter denoted as *T*1, *T*2, *T*3, and *T*4) and 4 of non-malignant (denoted as *N*1, *N*2, *N*3, and *N*4) regions were collected from a surgically removed 32.0 mm pathological stage pT3NOMx [48] PTC tumor of a 33 year old Asian woman. Tumor gene expression data [49] were compared with those from 4 papillary BCPAP [50] (denoted as: Φ1, Φ2, Φ3, and Φ4) and 4 anaplastic 8505C [51] (denoted as: Θ1, Θ2, Θ3, and Θ4) thyroid cancer cell line culture dishes [52]. The 8505C cell line was chosen for comparison to check whether the expression profiles of cell lines derived from differentiated thyroid tumors (like BCPAP and TPC1 [53]) evolved in vitro into profiles closer to undifferentiated anaplastic thyroid tumors [54]. All thyroid cancer samples were profiled by us using an Agilent-026652 Whole Human Genome Microarray 4x44K v2 [55]. The wet protocol and the raw data are fully described in the publicly accessible GEO deposits [49,52].

### 2.2. Transcriptomic Analyses

The microarray data were filtered (eliminated all spots with foreground fluorescence less than twice in the background in one microarray), normalized (to the median of valid spots), and analyzed using our standard GFP algorithms (presented in [56]). In addition to the primary characteristics of adequately profiled individual genes in each condition of AVE, REV, and COR (Appendix A), we also considered the derived characteristics of Relative Expression Control (REC) and Coordination Degree (COORD) (Appendix B). 

The AVE, REV, COR, REC, and COORD alterations in the malignant region of the tumor were quantified according to the algorithms from Appendix C and averaged for the genes included in the selected KEGG-constructed functional pathways. The arbitrarily introduced absolute fold-change cut-off (e.g., 1.5×) to consider a gene as significantly regulated might be too strict for very stably expressed genes or too lax for the very unstably expressed ones across biological replicas. Therefore, we computed the absolute fold-change cut-off for each quantified gene considering the combined contributions of the biological variability in the compared conditions and the technical noise of the probing microarray spots [26]. 

Nevertheless, the transcriptomic distance (i.e., the Euclidian distance in the 3D orthogonal space of transcriptomic changes (WIR, ΔREC, and ΔCOORD) with respect to the normal tissue (defined in C6) is the most comprehensive measure of the cancer-induced transcriptomic alteration of the thyroid. 

### 2.3. Functional Pathways

We analyzed the following KEGG-constructed functional pathways responsible for the Genetic Information Processing in *homo sapiens* (hsa).

#### 2.3.1. Transcription

(POL) hsa03020 RNA polymerase [57]. 

(BTF) hsa03022 Basal transcription factors [58].

(SPL) hsa03040 Spliceosome [59].

#### 2.3.2. Translation

(RIB) hsa03010 Ribosome [60].

(AMI) hsa00970 Aminoacyl-tRNA biosynthesis [61].

(NCT) hsa03013 Nucleocytoplasmic transport [62].

(SUR) hsa03015 mRNA surveillance pathway [63].

(RBE) hsa03008 Ribosome biogenesis in eukaryotes [64].

#### 2.3.3. Folding, Sorting, and Degradation

(PEX) hsa03060 Protein export [65].

(PPE) hsa04141 Protein processing in endoplasmic reticulum [66].

(SIV) hsa04130 SNARE interactions in vesicular transport [67].

(UMP) hsa04120 Ubiquitin mediated proteolysis [68].

(SRS) hsa04122 Sulfur relay system [69].

(PRO) hsa03050 Proteasome [70].

(RND) hsa03018 RNA degradation [71].

#### 2.3.4. Replication and Repair

(DER) hsa03030 DNA replication [72].

(BER) hsa03410 Base excision repair [73].

(NER) hsa03420 Nucleotide excision repair [74].

(MIR) hsa03430 Mismatch repair [75].

(HOR) hsa03440 Homologous recombination [76].

(NHJ) hsa03450 Non-homologous end-joining [77].

(FAP) 03460 Fanconi anemia pathway [78].

#### 2.3.5. Chromosome

(ACM) hsa03082 ATP-dependent chromatin remodeling [79].

(PRC) hsa03083 Polycomb repressive complex [80].

In addition, we have analyzed, in the surgically removed PTC tumor, the regulation of the gene modules responsible for the cancer cells’ survival and proliferation included in KEGG-constructed hsa05200 pathways in cancer (PAC [81]).

## 3. Results

### 3.1. Independence of the Three Transcriptomic Characteristics of Individual Genes

Figure 1 shows the independence of the primary characteristics AVE, REV, and COR of the 30 quantified DER genes in the four types of samples: N, T, Φ, and Θ. Correlations with *PCNA* (proliferating cell nuclear antigen) were selected for the encoded protein’s role in ensuring the rate and accuracy of DNA replication, damage repair, chromatin formation, and segregation of the sister chromatids [82]. The characteristics were determined, with the definition algorithms described in Appendix A. 

Using the three characteristics (whose independence is visually evident) increases the amount of workable transcriptomic information that can be derived from any microarray (or RNA-sequencing) experiment by almost four orders of magnitude. Thus, by quantifying 14,904 unigenes in this experiment, we obtained for each type of sample 14,904 AVEs + 14,904 REVs + 111,057,156 CORs = 111,086,964 (i.e., 7453.5× more values to be analyzed than the traditional analysis limited to the 14,904 AVEs).

Interestingly, with respect to N, the cancer increased the average expression of these genes with 46% in T, 142% in Φ, and 146% in Θ. *RNASEH2A* (ribonuclease H2, subunit A), known for its role in protecting the genomic integrity and progression of prostate cancer [83], had the most remarkable increase in the cultured Φ (39.42×) and Θ (49.10×) cells. Although, to a much lower extent, expression of *RNASEH2A* was also significantly increased in T (1.96×). 

On the other hand, the large 91% REV of *RNASEH2B* (ribonuclease H2, subunit B) in the Θ cells indicates the high expression adaptability of this gene to ensure the proper DNA replication in any environmental conditions. Therefore, at least for the studied ATC cells, *RNASEH2B* may have little value, as a DNA damage response to targeted treatment [84]. 

Correlation analysis revealed that within the DER pathway, *PCNA* has 5 statistically significant synergistically expressed partners in N, but 11 in T, 21 in Φ, and 16 in Θ. These results indicate a substantial increase in synchronous expressions of DNA-replication genes in thyroid cancer. The numbers of antagonistically expressed partners are: zero in N, two in T, three in Φ, and zero in Θ. 

### 3.2. Cancer-Induced Regulation of Gene Expression Profile

#### 3.2.1. Measures of Regulation of Expression Level

Figure 2 presents three ways to quantify the cancer-related regulation of the expression level of 51 randomly selected out of 74 quantified genes encoding polymerases based on the algorithms described in Appendix C. 

The traditional measure of the percentages of significantly up- and down-regulated genes (according to a more or less arbitrarily introduced criterion) considers only the regulated genes and as uniform +1 or −1 contributors to the transcriptomic alteration. Instead, the other two measures from Figure 2 are applied to all genes, while also discriminating their contributions to the overall transcriptomic regulation. 

For instance, *POLR1C* (polymerase (RNA) I polypeptide C, 30 kDa) had the largest up-regulation in both cell culture lines even as its expression level was not significantly changed in the malignant part of the surgically removed tumor (*x^(T)^* = −1.09; *x^(^*^Φ*)*^ = 198.71; *x^(^*^Θ*)*^ = 221.73). *POLR1C* was reported as one of the ten most up-regulated proteins when infiltrative gastric cancer regions were compared to the adjacent normal tissue [85]. *TAF1A* (TATA-box binding protein associated factor, RNA polymerase I subunit A) was also a highly up-regulated gene in both cell lines, although it was significantly down-regulated in the tumor (*x^(T)^* = −2.12; *x^(^*^Φ*)*^ = 105.70; *x^(^*^Θ*)*^ = 128.88).

From the WIR perspective (Figure 2c), the largest negative contribution to the transcriptomic alteration of the polymerases were given in the two cell lines by *PTRF* (polymerase I and transcript release factor), although it appeared as not being significantly regulated in the tumor. The difference between high fold-change but low WIR for *POLR1C* and low absolute fold-change but high absolute WIR for *PTRF* in both cell lines comes from their difference in the average expression levels in the normal tissue: *AVE_POLR1C_* = 0.17, *AVE_PTRF_* = 16.80 (*AVE* is measured in expression levels of the median gene for that condition). *POLR2L* (RNA polymerase II, I, and III subunit L) is another polymerase gene with very large negative contributions to the BCPAP and 8505C transcriptomes owing to its large expression level (*AVE_POLR2L_* = 18.11) in the normal tissue.

#### 3.2.2. Regulation of the KEGG-Constructed Functional Pathways Responsible for the Genetic Information Processing in the Malignant Region of the Thyroid Tumor 

The regulated genes listed in Table 1 were identified using the cut-off criteria defined in Appendix C (A7) and the software “#PATHWAY#” (Version 1) described in Ref. [46]. 

#### 3.2.3. Regulation of the Protein Processing in Endoplasmic Reticulum Pathway

Figure 3 presents the localization of the regulated genes from the hsa04141 KEGG-constructed PPE pathway Protein Processing in Endoplasmic Reticulum [66] in the surgically removed PTC tumor. Within this gene subset, *CRYAB* (crystallin, alpha B), whose silencing might occur at the end of a stepwise dedifferentiation process in the thyroid gland [86], had the largest negative contribution in both PTC types of samples (x^(T)^ = −7.16, WIR^(T)^ = −105; x^(Φ*)*^ = −36.62, WIR^(Φ*)*^ = −610).

#### 3.2.4. Regulation of the Cancer Cells’ Survival and Proliferation Genes

Figure 4 presents the localization of the regulated genes responsible for the cancer cells survival and proliferation within the KEGG-constructed cancer functional pathways.

Genes in Figure 4 were identified by KEGG Pathways in Cancer [81] as responsible for blocking the differentiation, evading apoptosis, ensuring immortality and insensitivity to anti-growth signals, resistance to chemotherapy, as well proliferation and sustained angiogenesis. There is some degree of similarity between results in Figure 4 and those comparing gene expression profiles in the primary cancer nodule and the surrounding normal tissue from a surgically removed prostate tumor ([87], Figure S1a). For instance, *GSTO2* and *PGF* were down-regulated while *CSF1R, DDB2, HMOX1* and *PGFB* were up-regulated in both cancers. However, *GSTP1* and *PPARG* were down-regulated in PTC but up-regulated in the profiled prostate tumor. 

### 3.3. Additional Measures of Transcriptomic Alterations

Figure 5 presents our original novel measures of the cancer-induced transcriptomic alterations in expression control (ΔREC) and coordination degree (ΔCOORD), as well as the most comprehensive measure, the Transcriptomic Distance (TD). The measures, determined using the algorithms from Appendix C, were applied to the quantified 51 RNA polymerase II genes and their binding partners in N, T, Φ, and Θ.

Within this gene subset, the expression control (Figure 5a) had the largest increase for *TAF7* (TATA-box binding protein associated factor 7) in *T* (ΔREC = 126), *TAF6* (TATA-box binding protein associated factor 6) in Φ (ΔREC = 218), and *POLR2B* (RNA polymerase II subunit B) in Θ (ΔREC = 485). By contrast, the expression controls of *POLR2L* in *T* (ΔREC = −144) and Θ (ΔREC = -198), and *GTF2E1* (general transcription factor IIE subunit 1) in Φ (ΔREC = −166), were substantially diminished. Interestingly, while the expression control of *TAF6* increased in Φ, it stayed practically the same in *T* (ΔREC = 5), but decreased in Θ (ΔREC = −46), indicating significant changes in cancer cells’ priorities for controlling the expression fluctuations of this gene. The increased expression control of *TAF7* is justified by its essential role for transcription, proliferation, and differentiation [88].

The coordination degree (Figure 5b) was also largely affected by cancer. The largest increase was exhibited by *ERCC3* (excision repair 3 TFIIH core complex helicase subunit) in *T* (ΔCOORD = 34) and Θ (ΔCOORD = 64), and *TAF1L* (TATA-box binding protein associated factor 1 like) in Φ (ΔCOORD = 60). Expression coordination of *ERCC3*, also known as xeroderma-pigmentosum B (XPB) [89], also increased in Φ (ΔCOORD = 32). In contrast, *GTF2I* (general transcription factor IIi) in *T* (ΔCOORD = −44), *GTF2H4* (general transcription factor IIH subunit 4) in Φ (ΔCOORD = −46), and *TAF6* (TATA-box binding protein associated factor 6) in Θ (ΔCOORD = −44) exhibited the largest decrease. 

By the most comprehensive measure (Transcriptomic Distance, TD (Figure 5c)), *POLR2L* had the largest contribution to the overall cancer-induced transcriptomic alteration within this gene subset in *T* (***TD^(T)^* = 145**, *TD^(^*^Φ^*^)^* = 497, *TD^(^*^Θ*)*^ = 495) and *GTF2I* in both cell lines (*TD^(T)^* = 45, ***TD^(^*^Φ^*^)^* = 964** *TD^(^*^Θ*)*^ = **872**). The least affected genes were: *POLR2H* (RNA polymerase II subunit H) in *T (**TD^(T)^***
**= 3**, *TD^(^*^Φ^*^)^* = 106, *TD^(^*^Θ*)*^ = 157), *TBP* (TATA-box binding protein) in Φ *(TD^(T)^* = 15, ***TD^(^*^Φ^*^)^* = 26**, *TD^(^*^Θ*)*^ = 18), and *POLR2D* (general transcription factor IIH subunit 1) in Θ *(TD^(T)^* = 15, *TD^(^*^Φ^*^)^* = 106, *TD^(^*^Θ*)*^ = 157). The average TD of this gene subset was 40 for the *T* samples, 99 for the BCPAP cells, and 94 for the 8505C cells. It is important to note the substantially different contribution levels to the overall transcriptomic change of this gene subset in the three types of samples.

### 3.4. Cancer-Induced Remodeling of DNA Replication (DER) Pathway 

Figure 6 presents the statistically (*p* < 0.05) significant synergism (COR > 0.950), antagonsim (COR < −0.950), and independence (|COR| < 0.050) of the expressions of genes involved in the sequentially connected multistep leading and lagging strands of the DNA replication [72]. 

### 3.5. Remodeling of the Coupling of Polymerase II Genes with Basal Transcription Factors

Figure 7 presents the statistically (*p* < 0.05) significant synergism and antagonism of the expressions of the polymerase II genes with their binding partners in the four types of the profiled samples. Figure 7 also indicates the genes within this selection that were found to be statistically (*p* < 0.05) significantly regulated in the cancer samples with respect to the normal tissue.

## 4. Discussion

A cancer patient is not a statistic entity, but a unique individual. Although cancer cells proliferate uncontrollably in every diseased person, the detailed molecular mechanisms come in unlimited flavors regulated by favoring factors whose dynamic combination is unique to each human. Moreover, the response to the favoring factors is not deterministic, but stochastic in nature. Therefore, meta-analyses of large populations disregard the personal specificity, being able to show only whether the distributions of outcomes is biased among races, sexes, age groups, and other general common factors, including some most frequently mutated genes like *BRAF* [90]. What happens to a particular gene is not important, since its regulation is not only different from person to person, but even among histo-pathologically distinct regions of the same tissue, as shown in many publications (e.g., [91,92]). We have also proved the transcriptomic heterogeneity by profiling prostate [87,93] and kidney [26,56] tumors. Regardless of what happens to individual genes, what is important is the alteration of the functional pathway as a whole and the consequences for the proliferation of the cancer cells and invasion of other tissues. Such alteration can occur not only through regulating the expression level of the composing genes, but also through the remodeling of their networking and changing the differential homeostatic control of their allowed fluctuations. 

As argued in a recent Editorial [94], there is no real cancer gene biomarker, since almost all genes were reported as mutated or/and regulated in all types of cancer cases. There is also no fit-for-all cancer gene therapy, since not only each person, but even each clone in the same tumor has a distinct response to the manipulation of a particular gene. Therefore, a personalized approach is needed. Meta-analyses of large populations can only validate the procedure to characterize the remodeling of functional pathways, nor the statistical relevance of a particular mutation or regulation of a given gene. On this line, this study presents the ways that thyroid cancer alters the genetic information processing pathways in one surgically removed tumor and two standard cell lines. Previously, we have reported significant alteration of the cell cycle, thyroid hormone synthesis, and oxidative phosphorylation pathways in a PTC tumor [25]. 

As standard in our laboratory [93] and also adopted by several other laboratories (e.g., [95,96,97]), the surrounding normal tissue in the tumor is the best reference to understand the genomic changes in the cancer nodule(s). Moreover, the purpose of any anti-cancer therapy is not to transform the tumor into the abstract average healthy human, but to restore the normality of the current patient’s tissue. Repeating the investigation on other patients/cell lines will only show that the genetic information processing is altered, but the concrete alterations of individual genes would be most likely different.

In this report, we used the GFP algorithms [45] to reveal the novel characteristics of the transcriptome that are ignored by the traditional analysis limited to the genes’ expression levels (e.g., [16,98]). Adding the independent REV and CORs characteristics to the AVE of each individual genes increased the information obtained from our gene expression experiments on thyroid cancer samples by almost 75 hundred times [49,52]. The independence of the three characteristics, presented here for 30 DNA replication genes in all four types of samples, was confirmed for several other gene sets profiled on surgically removed tumors from the kidney [26,56], prostate [87,93], and thyroid [25]. Altogether, AVEs, REVs, and CORs of all genes characterize the transcriptome as complete, as the numbers (AVE) of electronic devices of each type, their wiring (COR), and applied limits to the voltage fluctuations (REV) characterize a supercomputer. By evidence, only knowing the numbers of electronic devices of each type is not enough to build the supercomputer, since there are almost infinite possibilities to wire and subject them to voltage oscillations.

REV (Equation (A3)) and COR (Equation (A4)) were used to define the derived characteristics of individual genes REC (Equation (A5)) and COORD (Equation (A6)). The derived characteristics were used to define the novel measures of transcriptomic alteration ΔREC (Equation (A10)) and ΔCOORD *(*Equation (A11)). 

It is important to specify that the arbitrarily introduced (1.5× or 2.0×) absolute fold-change cut-off to consider a gene as significantly regulated was replaced by the flexible CUT (Equation (A7)). CUT is computed for each transcript by evaluating the biological variability of the expression level across biological replicas and the technical noise of the probing spots in the compared conditions. Also important is that the expression levels of adequately quantified genes were normalized to the median expression of all genes in that condition, which improves the accuracy of the conditions’ comparisons. 

Instead of the traditional percentages of up- and down-regulated genes that is limited to the significantly regulated, implicitly considering every affected gene as a uniform +1/−1 contributor to the expression profile alteration, we use the Weighted Individual (gene) Regulation (*WIR*; Equation (A8)). *WIR* takes into account the total change of the expression level and the statistical confidence of the regulation. Altogether, *WIR*, *ΔREC*, and *ΔCOORD* were incorporated into *TD* (Equation (A12)), the most theoretically possible comprehensive measure of the cancer-related transcriptomic change. 

In the presented analyses, the transcriptomic regulations in the cancer samples *T*, Φ, and Θ were referred to the gene expression profile of the surrounding normal tissue in the surgically removed PTC tumor. While such comparison is natural for the cancer nodule, it might be disputable for the cell lines because of the differences in the cellular environment. In separate studies, we proved the profound remodeling of the transcriptomes of each of two cell types (mouse cortical astrocytes and immortalized precursor oligodendrocytes) when co-cultured in insert systems [99,100]. These results suggest that a homo-cellular culture does not exactly repeat the genomic properties of the main cell type in the tissue. Moreover, a comparison of the gene expressions in cultured PTC cells and anaplastic thyroid cancer (ATC) cells indicated that, after several passages, the PTC cells profile evolves closer to that of the ATC cells [54]. 

Nevertheless, although the (2–6 mm^3^) tissue samples were collected from the most apparently homogeneous regions of the surgically removed thyroid tumor, it is no guaranty that blood and immune cells were not present. Thus, more than reflecting normal and cancer thyrocytes, the recorded transcriptomic profiles are actually weighted averages of the transcriptomes of several types of cells, which is one substantial imitation of our work. 

The functional pathways analyzed in Table 1 are not mutually exclusive. For instance, *POLD4* (polymerase (DNA-directed), delta 4, accessory subunit) is included in hsa03030 DNA replication, hsa03410 Base excision repair, hsa03420 Nucleotide excision repair, hsa03430 Mismatch repair, and hsa03440 Homologous recombination. Interestingly, while we found *POLD4* as significantly up-regulated (x = 1.90) in the malignant part of the tumor compared to the normal region, it was massively down-regulated in both BCPAP (x = −26.96) and 8505C (x = −39.64) cells. A recent meta-analysis of TCGA (The Cancer Genome Atlas) data [101] has shown that *POLD4* was significantly overexpressed in 17 types of cancer compared to the adjacent normal tissue. Our results again indicate the limitations of the standardized cancer cell lines to replicate the characteristics of the real tumors, owing to the potential in vitro evolution and genomic instability caused by repeated passages of the cultured cells [54]. 

Figure 3 shows how the expressions of genes involved in the protein processing in endoplasmic reticulum were regulated in the cancerous part of the thyroid tumor. Interestingly, the ubiquitination gene *UBXN1* (UBX domain protein 1), an important negative regulator of the unfolded protein response [102], was significantly up-regulated in the profiled tumor (*x^(T)^* = 1.75), but massively down-regulated in both cell lines (*x^(^*^Φ*)*^ = −41.96, *x^(^*^Θ*)*^ = −106.57). While depletion of *UBXN1* might be profitable for cancer cell survival and proliferation through protecting against endoplasmic reticulum (ER) stress [103,104], its up-regulation might facilitate migration and invasion of the cancer cells within the thyroid tissue, as reported in prostate cancer [105]. Therefore, we speculate that, although in apparent contradiction, both *UBXN1* up-regulation in the tissue (Figure 3) and *UBXN1* down-regulation in the homo-cellular BCPAP and 8505C cultures are justified by the molecular mechanisms controlled by the encoded protein. *PTRF* and *UBE2O* (ubiquitin-conjugating enzyme E2O) are other examples of genes that were not affected in the tumor but significantly regulated in both BCPAP (*x_PTRF_ = −*31.50, *x_UBE2O_* = 28.86) and 8505C (*x_PTRF_ = -58.42*, *x_UBE2O_* = 11.70) cell lines. *UBE2O* was reported to ubiquitinate *PTRF* and inhibit the “secretion of exosome-related PTRF/CAVIN1” [106]. 

Figure 4 justifies the increased proliferation of the cancer cells through up-regulated *CCND1* (*x^(T)^* = 3.48; *x^(^*^Φ*)*^ = 4.78; *x^(^*^Θ*)*^ = 8.85), one of the 20 hub genes used as a biomarker for thyroid cancer [107]. Survival of the cancer cells is optimized through the up-regulation of several key genes, ensuring evading apoptosis such are *BIRC5* (*x^(T)^* = 6.04) and *DDB2* (*x^(T)^* = 2.37). *BIRC5* (*x^(^*^Φ*)*^ = 10.70; *x^(^*^Θ*)*^ = 7.35) is known for promoting multidrug resistance to chemotherapy [108], and the oncogenic roles of *DDB2* were reported in a recent review [109]. However, with respect to the normal thyroid tissue, *DDB2* was significantly downregulated in both cell lines (*x^(^*^Φ*)*^ = −3.01; *x^(^*^Θ*)*^ = −22.00). Within a hetero-cellular tissue (composed of normal and cancer thyrocytes and potential presence of immune cells), over-expression of genes such is *DDB2* accelerates cancer cell proliferation and invasion of the thyroid. Nevertheless, expression of genes that are no longer necessary is diminished in homo-cellular cultures [54]. 

The study proved that regulation of the gene expression profile is only a part of the global cancer-induced transcriptomic alteration, as shown in Figure 5 for 51 RNA polymerase II genes and their binding partners. We believe that changes in the control of genes’ expression fluctuations and their inter-coordination should also be considered and eventually incorporated into the more general measure like the Transcriptomic Distance.

On average, REC increased in *T* by 2%, in Φ by 28%, and in Θ by 20%. An increased control limits the fluctuations of the expression level within a narrow interval, while a decreased control allows the genes to easily adapt their expression levels to various environmental conditions. We believe that the Relative Expression Control is an indirect measure of cell’s priorities to ensure the right expression levels of critical genes for its survival and proliferation. As such, the regulation of REC gives important information of how cancerization has changed the cells’ priorities. We found a huge increase in the expression control of *POLR2B* in both the anaplastic (ΔREC = 485) and the papillary (ΔREC = 206) thyroid cancer cell lines, but not in the tumor (ΔREC = 5). The differences might be explained by the potential effects of the encoded protein on the tumor growth as recently reported in glioblastoma [110]. On the other hand, the much stricter control of *TAF6* in the BCPAP cells while being looser in the 8505C cells and practically not affected in the surgically removed cancer specimen may indicate different anti-tumor molecular mechanisms [111]. It is interesting to note that *POLR2B* was up-regulated in both cell lines, but not in the tumor (*x^(T)^* = 1.15; *x^(^*^Φ*)*^ = 74.40; *x^(^*^Θ*)*^ = 80.06), while *TAF6* was down-regulated in both cell lines, but not in the tumor (*x^(T)^* = −1.02; *x^(^*^Φ*)*^ = −1.91; *x^(^*^Θ*)*^ = −3.24). Thus, although, for the investigated thyroid tissue, both *POLR2B* and *TAF6* might be used as housekeeping genes [112], they are not suitable as references for cultured thyroid cancer cells. 

For the gene set in Figure 5b, on average, the coordination degree decreased in *T* (ΔCOORD) = −7.76), but increased in the two cell lines (ΔCOORD) = 8.96 in Φ and ΔCOORD) = 16.80 in Θ). Increased coordination means more synchronization of the genes’ expressions with direct consequences on the pathway efficiency. By contrast, decreased coordination makes the pathway more flexible through letting genes fluctuate their expression levels more independently. It seems natural to assume that the transcription pathways need more flexibility to adapt to the tissue heterogeneity, while in a homo-cellular dish the pathway efficiency comes first.

The high contributions of *POLR2L* to the transcriptomic alteration in all three types of thyroid cancer samples, as measured with the Transcriptomic Distance (Figure 5c), indicate that it might be a good target for anti-thyroid cancer gene therapy, as was suggested for prostate cancer [113]. Interestingly, the main contribution to the TD separating the state of a gene in a cancer sample with respect to the normal thyroid tissue may come from any of the three types of individual gene characteristics. Thus, in the case of *POLR2L*, *TD^(T)^ =* 145 comes mainly from the regulation of the expression control (*ΔREC^(T)^ =* −142). In contrast, in the cell cultures, the large TDs (*TD^(^*^Φ*)*^ = 497; *TD^(^*^Θ*)*^ = 495) came from their WIRs (*WIR^(^*^Φ*)*^ = −487; *WIR^(^*^Θ*)*^ = −452), because of the substantial down-regulations (x*^(^*^Φ*)*^ = −27.91; x*^(^*^Θ*)*^ = −25.97). The differences between the tumor and the cell cultures indicate that, although *POLR2L* is a major player in the development of several forms of thyroid cancers, the involved molecular mechanisms are distinct. Therefore, the therapy targeting this gene should be tailored to the specificity of the cancer form and patient personal characteristics.

Figure 6 and Figure 7 show how cancer remodels the gene network responsible for the DNA Replication and the coupling of the Polymerase II complex genes to the basal transcription factors in the investigated three types of thyroid malignancy. Expression levels of synergistically expressed genes fluctuate in the phase (simultaneously going up and down) across the biological replicas, while those of the antagonistically expressed fluctuate in the anti-phase (when one goes up the other goes down). Thus, COR analysis indirectly measures the expression synchronization of the coordinated genes. On the other hand, expression independence tells us that there is most likely no direct interaction of the encoded products by the two genes. Missing continuous red/blue or dashed black lines between the two genes does not mean that they do not interact, but that their expression correlation is not statistically significant in that condition. In addition to the substantial yet different remodeling of the transcriptomic network, it is also important to note the very different regulations of the gene expression profiles in the three types of thyroid cancer samples.

There are four main take home findings in Figure 6: (i)Cancer remodels the DNA replication pathway, as can be seen by comparing the gene network in the normal tissue (Figure 6a) and the malignant part (Figure 6b) of the same thyroid. Changes in the DER genes’ interaction might provide a complementary explanation for cancer development beyond the driving mutations [114].(ii)Expression correlation is independent of the regulation status of the linked genes. For instance, in “DPD →Final”, there is a significant synergism between the not-regulated *FEN1* and *REC1* in the PTC nodule of the tumor ((COR = 0.978; (Figure 6b)), but also between the up-regulated *FEN1* and the down-regulated *REC1* in the BCPAP cells (COR = 0.951; Figure 6c). The *FEN1–REC1* synergism was very near the 0.950 statistically significant cut-off in the normal tissue (COR = 0.944), but less close in the 8505C cells (COR = 0.749). FEN1 was reported to promote cancer cell proliferation, migration, and invasion [115] in biliary tumors.(iii)The remodeling depends on the form of thyroid cancer (see the differences between the papillary (Figure 6c) and anaplastic (Figure 6d) cell lines). For instance, the significant antagonism of the significantly down-regulated genes *RPA2* and *POLE4* in the BCPAP cells is switched into a significant synergism in 8505C cells (also present in the normal tissue). Although miR-519 was identified as a common upstream regulator of both *RPA2* and *POLE4* [116], our result might be the first confirmation of their direct interaction.(iv)Each affected person has a distinct DER gene network, as shown by the differences between the network in Figure 6b (PTC tumor collected from the thyroid of a 33y old Asian woman) and that in Figure 6c (PTC collected from a 76y old Caucasian woman [117]). For instance, the independently expressed *RPA2* and *REC4* in T are synergistically expressed in Φ (and also in Θ). This difference indicates the distinct molecular mechanisms of the DNA replication in the papillary cancer nodules of the two PTC donors which requires the personalized approach of anti-cancer therapy (e.g., [26,87,93]).

The general findings from Figure 6 can be also retrieved in Figure 7: (i)Cancer remodels the significant expression coordination of polymerases with their targeted partners (compare the correlations in the normal tissue (Figure 7a) and the malignant part (Figure 7b) of the same thyroid.(ii)Expression correlation is independent of the regulation status of the linked genes. For instance, *POLR2H*, a prognosis gene for the lung squamus cell carcinoma [118] and *ERCC3* are synergistically expressed in both T and Φ samples, although *POLR2H* was up-regulated in T but down-regulated in Φ.(iii)The remodeling depends on the form of the thyroid cancer (see differences between the papillary (Figure 7c) and anaplastic (Figure 7d) cell lines).(iv)Each affected person has distinct gene networks, as shown by the differences between the correlations in the two PTC type samples T and Φ. For instance, *POLR2H* and *CDK7* are synergistically expressed in T, but are antagonistically expressed in Φ.

## 5. Conclusions

By providing the most theoretically possible comprehensive measure of the transcriptome alteration in cancer, the GFP approach is the alternative of choice to the gene biomarker paradigm. However, GFP findings should be completed by better equipped laboratories through proteomics experiments to validate and quantify at the protein level the remodeling of the genetic information processing pathways identified by us at the transcript level. This is not an easy task since both the transcriptomic and the proteomic experiments should be carried on samples collected from the same hetero-cellular regions of the same tumor. Although investigating tumors from different persons would most likely identify distinct regulations of individual genes, the overall correlation between the transcriptomic and the proteomic alterations might remain. Thus, the value of the present study resides mainly in the methodological avenue towards the personalized approaches of cancer gene diagnosis and therapy. 

## Figures and Tables

**Figure 1 genes-15-00621-f001:**
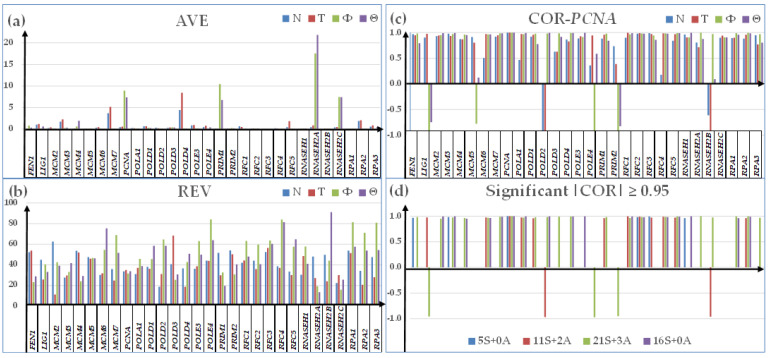
Independence of the three transcriptomic characteristics of the 30 quantified DNA-replication genes in the normal thyroid tissue (N), papillary thyroid cancer tissue (T), papillary thyroid cancer cell line BCPAP (Φ), and anaplastic thyroid cancer cell line 8505C (Θ). (**a**) Average expression level (AVE). (**b**) Relative Expression Variation (REV). (**c**) Expression correlation (COR) with PCNA (proliferating cell nuclear antigen). (**d**) Statistically (*p* < 0.05) significant correlations (|COR| > 0.95) of the DER genes with PCNA.

**Figure 2 genes-15-00621-f002:**
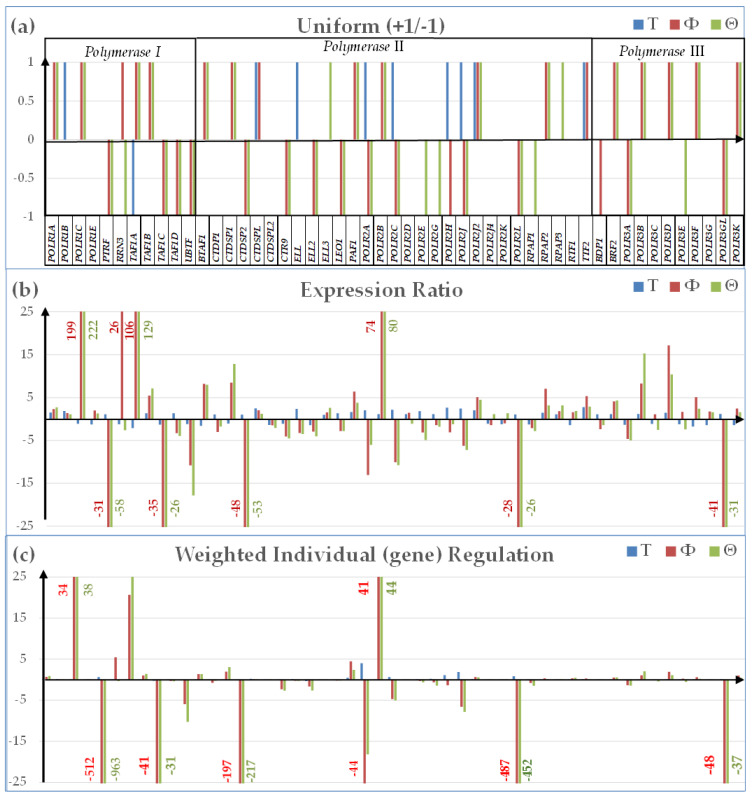
Three ways to measure the expression regulation of 51 randomly selected out of 74 quantified genes encoding polymerases in cancer: (**a**) Uniform (+1/−1) contribution; (**b**) Expression Ratio; and (**c**) Weighted Individual (gene) Regulation.

**Figure 3 genes-15-00621-f003:**
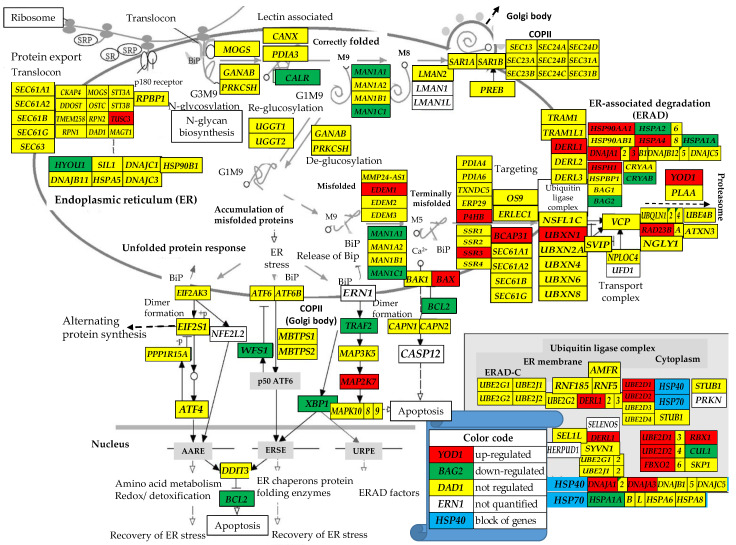
Modified from [66]: Regulation of the KEGG-constructed pathway Protein Processing in Endoplasmic Reticulum [66] in the surgically removed PTC tumor. BiP = Immunoglobulin Binding protein, SRP = signal recognition particle. Up-regulated genes: *BAX* (BCL2-associated X protein), *BCAP31* (B-cell receptor-associated protein 31), *DERL1* (derlin 1), *DNAJA1* (DnaJ heat shock protein family (Hsp40) member A1), *EDEM1* (ER degradation enhancing α-mannosidase like protein 1)*, FBXO2* (F-box protein 2), *HSP90AA1* (heat shock protein 90 α family class A member 1), *HSPH1* (heat shock protein family H (Hsp110) member 1), *MAP2K7* (mitogen-activated protein kinase kinase 7), *P4HB* (prolyl 4-hydroxylase subunit β)*, RAD23B* (RAD23 homolog B, nucleotide excision repair protein), *SSR3* (signal sequence receptor subunit 3), *TUSC3* (tumor suppressor candidate 3), *UBE2D1/2* (ubiquitin conjugating enzyme E2 D1/D2), *UBXN1* (UBX domain protein 1), and *YOD1* (YOD1 deubiquitinase). Down-regulated genes: *BAG2* (BCL2-associated athanogene 2), *BCL2* (B-cell CLL/lymphoma 2), *CALR* (calreticulin)*, CRYAB*, *HERPUD1* (homocysteine inducible ER protein with ubiquitin like domain 1), *HSPA1A* (heat shock protein family A (Hsp70) member 1A), *HYOU1* (hypoxia up-regulated 1), *MAN1A1/C1* (mannosidase α class 1A/1C member 1), *WFS1* (wolframin ER transmembrane glycoprotein), and *XBP1* (X-box binding protein 1).

**Figure 4 genes-15-00621-f004:**
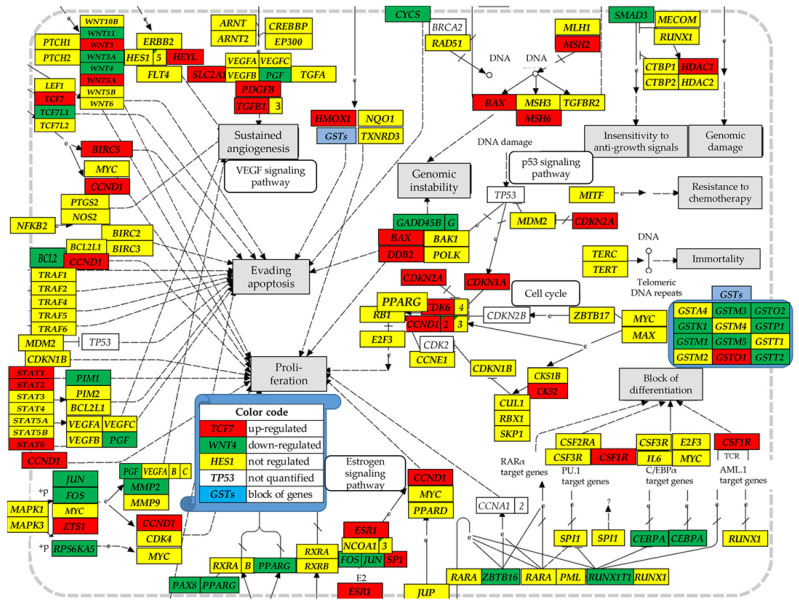
Modified frm [81]: Localization of the regulated genes responsible for the cancer cells survival and proliferation within the KEGG-constructed functional pathways in cancer for the surgically removed PTC tumor. Regulated genes: *BAX, BCL2*, *BIRC5* (baculoviral IAP repeat containing 5), *CCND1* (cyclin D1), *CDK6* (cyclin-dependent kinase 6), *CDKN2A* (cyclin-dependent kinase inhibitor 1A), *CEBPA* (CCAAT/enhancer binding protein (C/EBP), α), *CKS2* (CDC28 protein kinase regulatory subunit 2), *CSF1R* (colony stimulating factor 1 receptor), *DDB2* (damage-specific DNA binding protein 2), *ESR1* (estrogen receptor 1), *ETS1* (v-crk avian sarcoma virus CT10 oncogene homolog-like), *FOS* (FBJ murine osteosarcoma viral oncogene homolog), *GTSK1/M1/M3/M5/o1/O2P1/T2* (glutathione S-transferase kappa 1/mu 1/mu 3/mu 5/omega 1/omega 2/pi 1/theta 2), *HDAC1* (histone deacetylase 1), *HEYL* (hes-related family bHLH transcription factor with YRPW motif-like), *HMOX1* (heme oxygenase (decycling) 1), *JUN* (jun proto-oncogene), *MMP2* (matrix metallopeptidase 2 (gelatinase A, 72kDa gelatinase, 72kDa type IV collagenase)), *MSH6* (mutS homolog 3), *PAX8* (paired box 8), *PDGF* (platelet-derived growth factor β polypeptide), *PGF* (placental growth factor), *PIM1* (pim-1 oncogene), *PPARG* (peroxisome proliferator-activated receptor γ), *RUNX1T1* (runt-related transcription factor 1; translocated to, 1 (cyclin D-related)), *RPS6KA5* (ribosomal protein S6 kinase, 90kDa, polypeptide 5), *SLC2A1* (solute carrier family 2 (facilitated glucose transporter), member 1), *SMAD3* (SMAD family member 3), *SP1* (Sp1 transcription facto), *STAT1/2/6* (signal transducer and activator of transcription 1), *TCF7* (transcription factor 7 (T-cell specific, HMG-box)), *TCF7L1* (transcription factor 7-like 1 (T-cell specific, HMG-box)), *TGFB1* (transforming growth factor, β 1), *WNT11/3/3A/4/5A* (wingless-type MMTV integration site family, member 11/3/3A/4/5A), and *ZBTB16* (zinc finger and BTB domain containing 17).

**Figure 5 genes-15-00621-f005:**
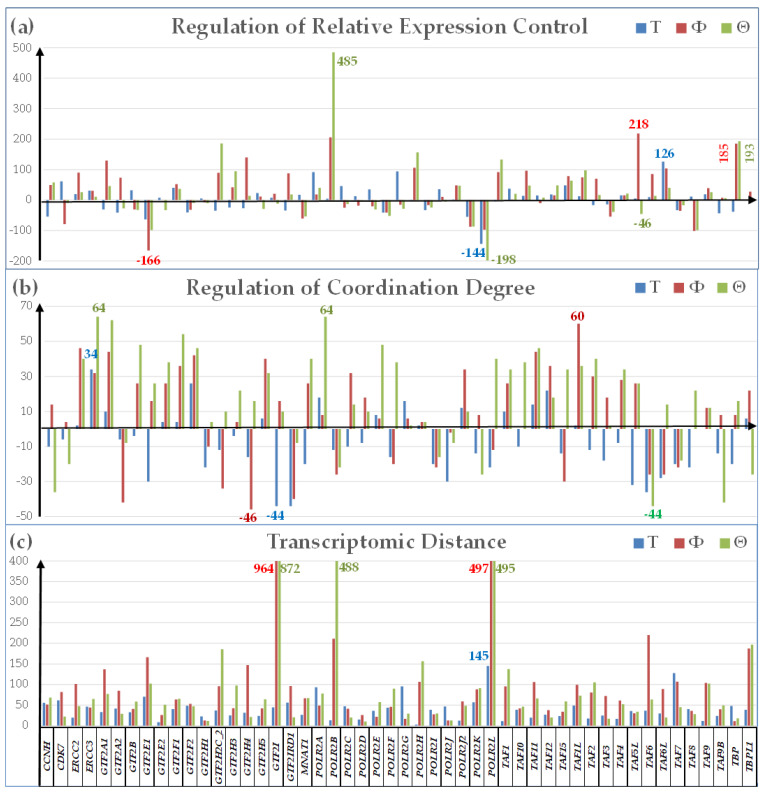
Novel measures of the transcriptomic alteration of 51 RNA polymerase II genes and their binding partners with respect to the normal thyroid tissue in the surgically removed PTC tumor (*T*), PTC cell line BCPAP (Φ), and APC cell line 8505C (Θ). (**a**) Regulation of the Relative Expression Control; (**b**) Regulation of the Coordination Degree; (**c**) Transcriptomic Distance.

**Figure 6 genes-15-00621-f006:**
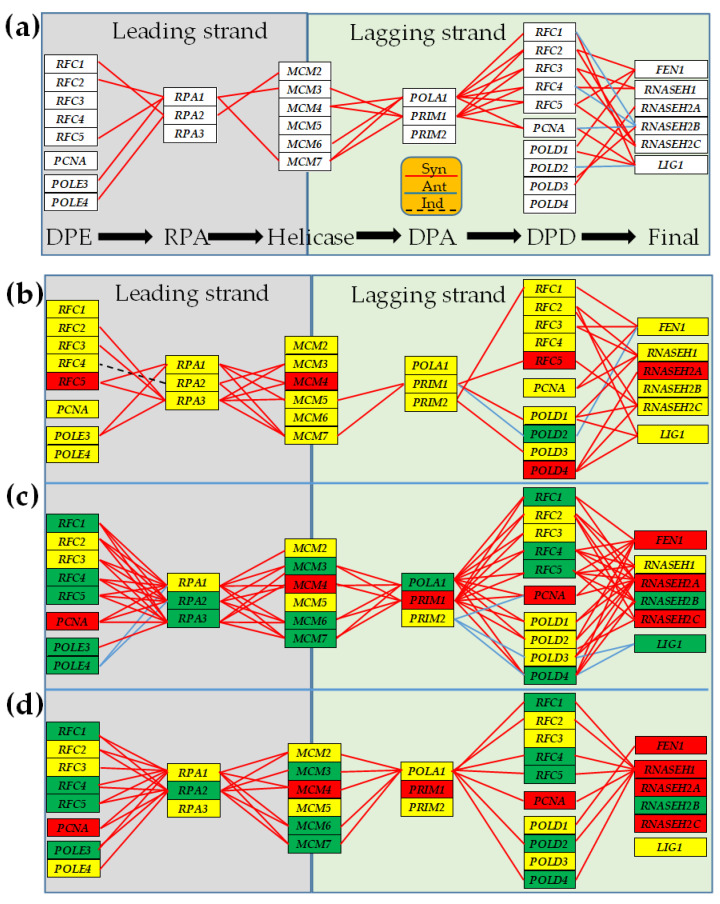
Statistically (*p* < 0.05) significant transcriptomic network of the KEGG-constructed DNA Replication Pathway in the: (**a**) normal thyroid tissue, (**b**) papillary thyroid cancer tissue, (**c**) papillary thyroid cancer cell line BCPAP, and (**d**) anaplastic thyroid cancer cell line 8505C. Continuous red lines indicate significant expression synergism (e.g., *RFC1–RPA1* in (**a**)). Continuous blue lines (e. *RPA1–POLE4* in (**c**)) indicate significant expression antagonism. Dashed black lines (e.g., *RPA2–RFC4* in (**b**)) indicate the significant independence of the two genes. A missing line means that the expression correlation was not (*p* < 0.05) statistically significant. The red/green/yellow background of the gene symbols indicates whether that gene was significantly up-/down-/not-regulated. Gene blocks: **DPE** = DNA polymerase ε complex; **RPA** = Replication proteins A; **Helicase** = MCM complex; **DPA** = DNA polymerase α-primase complex; **DPD** = DNA polymerase δ complex, **Final** = the end stage (RNaseHI–RNaseHII–Fen1–DNA ligase).

**Figure 7 genes-15-00621-f007:**
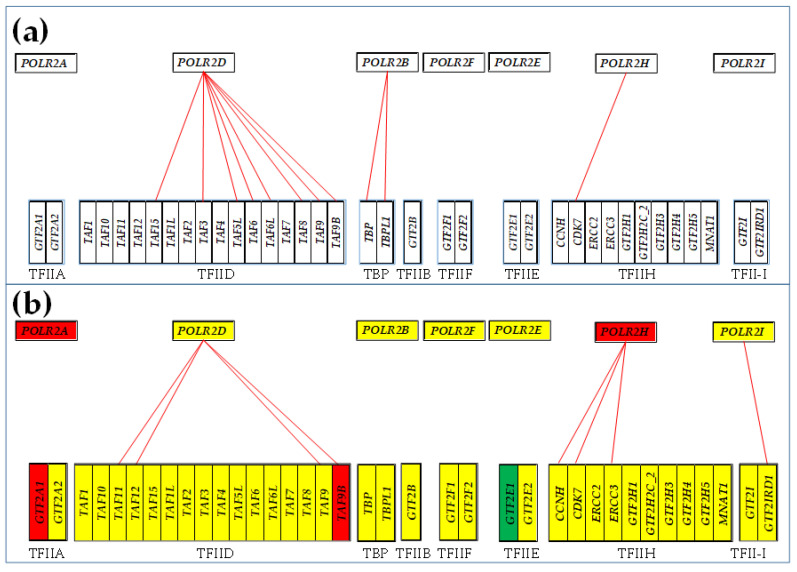
Statistically (*p* < 0.05) significant synergism and antagonism of the expressions of the polymerase II genes with their direct binding partners in the normal thyroid tissue (**a**), papillary thyroid cancer tissue (**b**), papillary thyroid cancer cell line BCPAP (**c**), and anaplastic thyroid cancer cell line 8505C (**d**). Note the differences in the significant coupling and the regulation of the composing genes among the four conditions.

**Table 1 genes-15-00621-t001:** Statistically significantly up- and down-regulated genes from the KEGG-constructed functional pathways responsible for the Genetic Information Processing in the malignant region of the thyroid tumor. Symbols of up-regulated genes are in bold letters.

TRANSCRIPTION	
RNA polymerase	***POLR1B***, ***POLR2A***, ***POLR2C***, ***POLR2H***, ***POLR2J***, ***POLR2J2***
Basal transcription factors	***GTF2A1***, ***TAF9B***, *GTF2E1*
Splicesome	***CDC40***, ***EFTUD2***, ***HNRNPA1***, ***HNRNPC***, ***LSM5***, ***PHF5A***, ***RBM8A***, ***RBMX***, ***SNRNP70***, ***SNRPD1***, ***SNRPG*** *BCAS2*, *HSPA1A*, *LSM6*, *PPIL1*, *PRPF40A*, *RNU2-1*, *RNVU1-18*, *SRSF6*
TRANSLATION	
Ribosome	***MRPL14***, ***MRPL21***, ***MRPS6***, ***RPL14***, ***RPL28***, ***RPL30***, ***RPLP1*** *RPL10A*, *RPL17*, *RPL18A*, *RPL21*, *RPL23*, *RPL26*, *RPL27*, *RPL31*, *RPL34*, *RPL35A*, *RPL6*, *RPS10*, *RPS12*, *RPS14*, *RPS16*, *RPS20*, *RPS25*, *RPS27*, *RPS3A*, *RPS5*, *RPS6*, *RPS7*, *RPS8*
Aminoacyl-tRNA biosynthesis	***IARS2***, ***NARS2***
Nucleocytoplasmic transport	***KPNA6***, ***NUP205***, ***NXF1***, ***NXF3***, ***RBM8A***, ***TNPO2*** *NUP153*, *NUP93*, *XPO4*
mRNA surveillance pathway	***NXF1***, ***NXF3***, ***PAPOLG***, ***PPP2CA***, ***PPP2R2D***, ***PPP2R3B***, ***RBM8A***, ***SMG6*** *PELO*, *PPP2R2B*
Ribosome biogenesis in eukaryotes	***DROSHA***, ***FBLL1***, ***HEATR1***, ***NHP2***, ***NXF1***, ***NXF3***, ***POP4***, ***RCL1***, ***RPP40*** *FBL*, *SNORD3B*
FOLDING, SORTING AND DEGRADATION	
Protein export	** *SRP9* **
Protein processing in endoplasmic reticulum	***BAX***, ***BCAP31***, ***DERL1***, ***DNAJA1***, ***EDEM1***, ***FBXO2***, ***HSP90AA1***, ***HSPH1***, ***MAP2K7***, ***P4HB***, ***RAD23B***, ***SSR3***, ***TUSC3***, ***UBE2D1***, ***UBE2D2***, ***UBXN1***, ***YOD1*** *BAG2*, *BCL2*, *CALR*, *CRYAB*, *HERPUD1*, *HSPA1A*, *HYOU1*, *MAN1A1*, *MAN1C1*, *WFS1*, *XBP1*
SNARE interactions in vesicular transport	***STX4***, ***VAMP1***, ***VAMP4***, ***VAMP8*** *STX11*, *STX1A*, *STX2*
Ubiquitin-mediated proteolysis	***CBL***, ***DDB2***, ***FBXO2***, ***HERC4***, ***KEAP1***, ***MAP3K1***, ***MGRN1***, ***RNF7***, ***UBA1***, ***UBB***, ***UBE2A***, ***UBE2C***, ***UBE2D1***, ***UBE2D2***, ***UBE2H*** *PPIL2*
Proteasome	***PSMA7***, ***PSMB1***, ***PSMB2***, ***PSMB3***, ***PSMB4***, ***PSMD14***, ***PSMD4***, ***PSMD6***, ***PSME4***
RNA degradation	***CNOT10***, ***ENO3***, ***EXOSC1***, ***EXOSC3***, ***LSM5***, ***PFKM*** *LSM6*
REPLICATION AND REPAIR	
DNA replication	***MCM4***, ***POLD4***, ***RFC5***, ***RNASEH2A*** *POLD2*
Base excision repair	***NEIL1***, ***PARP1***, ***PNKP***, ***POLD4***, ***RFC5*** *POLD2*, *POLG2*
Nucleotide excision repair	***DDB2***, ***POLD4***, ***POLR2A***, ***POLR2C***, ***POLR2H***, ***POLR2J***, ***POLR2J2***, ***RAD23B***, ***RFC5***, ***XPA*** *POLD2*
Mismatch repair	***MSH2***, ***MSH6***, ***POLD4***, ***RFC5*** *POLD2*
Homologous recombination	***POLD4***, ***RAD50***, ***XRCC3*** *POLD2*
Non-homologous end-joining	** *RAD50* **
Fanconi anemia pathway	***FANCE***, ***FANCI***, ***POLH***, ***RMI2*** *FAN1*
CHROMOSOME	
ATP-dependent chromatin remodeling	***ARID1A***, ***BAZ1A***, ***BAZ2A***, ***BCL7A***, ***BCL7B***, ***BCL7C***, ***HDAC1***, ***KAT5***, ***MEAF6***, ***PBRM1***, ***RSF1S***, ***MARCA4***, ***SMARCD3***, ***SMARCE1***, ***YEATS4*** *CECR2 ING3 MBD2*
Polycomb-repressive complex	***AEBP2***, ***CBX2***, ***CBX4***, ***EZH1***, ***HDAC1***, ***PHF19***, ***SCMH1***, ***TEX10***, ***UBE2D1***, ***UBE2D2***, ***YAF2*** *ASXL3*, *USP16*

## Data Availability

Gene expression data used in this study were downloaded from the publicly accessible websites: https://www.ncbi.nlm.nih.gov/geo/query/acc.cgi?acc=GSE97001 and https://www.ncbi.nlm.nih.gov/geo/query/acc.cgi?acc=GSE97002 (accessed on 4 February 2024).

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
