# Peer review of "Papillary Thyroid Cancer Remodels the Genetic Information Processing Pathways"

_genes, 2024, doi:10.3390/genes15050621_

Round 1

Reviewer 1 Report

Comments and Suggestions for Authors

COMMENTS 

The manuscript titled “Papillary Thyroid Cancer Remodels the Genetic Information Processing Pathways” of Iacobas DA et al., reports an investigation about the characterization of transcriptomes appearing in thyroid cancers through the Genomic Fabric Paradigm (GFP) perspective.

Two were the aims of this investigation:

1.      to characterize transcriptomes by reporting of each gene five features such as average expression level (AVE), relative expression variation (REV) across biological replicas, expression coordination (COR) with each other gene in the same condition, relative expression control (REC) and coordination degree (COORD).

2.      To identify the most legitimate targets for personalized anti-cancer gene therapy.

To perform this retrospective study, 8 small pieces of surgically removed thyroid tissues of a 33y old Asian woman were analyzed. Mainly, 4 of these pieces (T1, T2, T3, T4) were picked up from malignant region of PTC at pT3NOMx pathological stage. Conversely, the remain 4 pieces were collected from non-malignant regions (N1, N2, N3, N4).

By using Agilent-026652 Whole Human Genome Microarray 4x44K v2, these samples were profiled and then, compared with transcriptomic raw data of papillary thyroid cancer cell line BCPAP (Φ) and anaplastic thyroid cancer cell line 8505C (Θ) downloaded from the publicly available Gene Expression Omnibus (GEO) of the National Center for Biotechnology Information (NCBI). AVE, REV, COR, REC and COORD features were reported according to the GFP algorithms.

Abstract:

Abstract section is describing this study. However,

Minor:

lines 11-13: this sentence should be revised. I suggest to authors to avoid to write the results of their past investigation in “Abstract” section. Conversely, this is a good information to write in “Introduction” section.

Lines 13-16: this sentence should be clearly written for effective communication of aims and material and methods employed to achieve them.

Lines 16-17: BCPAP abbreviation is superfluous.

Lines 17-19: this sentence concerns the overall approach of investigation (to analyze transcriptomes of papillary thyroid carcinomas and anaplastic variants); therefore, should be placed before the specific methods to obtain the results of investigation (to compare profiles of two different thyroid cancers). Further, GFP abbreviation is superfluous.

Lines 19-22: probably, the Authors are considering “the results of study” indicated that… further, this complex sentence should be replaced by short sentences to make easier to understand.

Keywords:

It is better to associate words to abbreviations.

Introduction:     

This section is adequately describing the aims of study.

However, I have three comments for Authors about:

1.      lines 50-54: “the general question of the biomarkers’ utility for cancer diagnostic [18] and therapy [19]…” this sentence should be updated because this is not a general question. Indeed, this is a specific issue of the outmost importance since  clear international taxonomies from the U.S. Food and Drug Administration (FDA), and the European Medicines Agency (EMA) have already defined what should be included among biomarkers and their applications in management of thyroid cancers (doi.org/10.3390/diagnostics12030662).

2.      Lines 69-76: these three sentences should be part and parcel of “Discussion” section.

3.      Lines 83-99: these six sentences should be part and parcel of “Material and Methods” section.

Materials and Methods:           

This section provides sufficient information. However,

Minor:

lines 110-111: this sentence should be revised because there is a repeated word (Agilent).

Results:

This section provides detailed information. However,

Minor:

lines 170-173: this sentence should be simplified. Probably, “as showed in Figure 1” instead of “Figure 1 illustrates”. In the same line, lines 208-210, lines 246-248, lines 269-270, lines 304-306, lines 345-348, lines 363-364: “as showed in Figure…”

Line 183-184: is “It is interesting to note that” is superfluous. These are the Results of investigation. Further, “normal tissue” has been previously abbreviated as “N”.

Discussion:

The comments of discussion are appropriate for this investigation. Mainly, this section starts by a solid bases on which to build adequate interpretations of results.

A comment about biomarkers, diseases (cancers) and subjects suffering from (cancer). The biomarkers are made for diseases that are ideal entities. In fact, we describe diseases through numbers (incidence, dimension, recurrence) that show commas and therefore, they are out of reality. We use numbers and rules to justify biomarkers since simplifying we can better understand. Your investigation is a good tempt to overcome this gap. The subject suffering from cancer is a complex unicum who knows even as to keep at bay the disease. PTC is a good example when it is an incidental find during autopsy.

Minor:

lines 433-436: several words have been previously abbreviated. Conversely, in lines 436-437 the abbreviations are not reported.

Line 485: probably, could be simpler to give the answer without writing the question.

Lines 533-536: underline text should be avoided.

Conclusions:     

The conclusions are relevant.

Table and Figures: give a helpful visual representation of study.

References:

References are appropriate.

Decision:

This study may be accepted for publication after minor revisions.

Comments on the Quality of English Language

No comments

Reviewer 2 Report

Comments and Suggestions for Authors

In the study, the Authors determine how changes in the transcriptome affect the DNA replication, repair and transcription. The study is interesting and well-written. Unfortunately, the manuscript cannot be accepted in the current form. Suggested changes are listed below.

[1]. I wonder why a cell line derived from poorly differentiated thyroid carcinoma (i.e., cell line 8505C) was chosen, when only results for PTC tumor are included in the study. Perhaps it would have been more reasonable to use two different lines representing PTC (e.g. BCPAP and TPC1?). Please explain.

[2]. Lines 62-64: A sentence describing targeted therapies used to treat thyroid cancer should be corrected. It says that dabrafenib is a multi-kinase inhibitor, which is not true. This inhibitor (like trametinib, for example) belongs to specific kinase inhibitors (SKIs). This topic is further explained in the following publication, which I would suggest to cite here: Int J Mol Sci. 2021 Oct 31;22(21):11829. doi: 10.3390/ijms222111829.

[3]. Please write gene names in italics.

[4]. Please change “hsa05200Pathways” for “hsa05200 Pathways”.

[5]. Please change “BPAP” for “BCPAP” in lines 434, 443, 456, 472, 475 and 505. Similarly, replace   “X505C” with “8505C”.

[6]. Remove the underlines used in lines 553 and 555.

Author Response

Please see the attached point-by-point response

Reviewer 3 Report

Comments and Suggestions for Authors

I think it is an interesting study with statistically significant results. However, the authors report that p<0.95 is statistically significant: they should correct it and should indicate p<0.05 as a statistically significant value. Additionally, references should be updated with more recent literature data, especially in the introduction, deleting references older than 10 years and replacing them with more recent ones (such as: https://doi.org/10.1007/s12325-020-01391-1; https://doi.org/10.1007/s12020-022-03146-0; https://doi.org/10.1007/s12325-020-01391-1; doi: 10.1002/cncy.22454). I also think that the authors should better discuss the limitations of their work. Moreover, a thorough editing of English and a spelling and punctuation check should be performed.

Comments on the Quality of English Language

English language must be ameliorated.

Author Response

Thank you for the kind appreciation and valuable suggestions. Attached, please find the point-by-point our responses.
